# Neuroprotection of Kaji-Ichigoside F1 via the BDNF/Akt/mTOR Signaling Pathways against NMDA-Induced Neurotoxicity

**DOI:** 10.3390/ijms232416150

**Published:** 2022-12-18

**Authors:** Faju Chen, Li Wang, Fengli Jin, Liangqun Li, Tao Wang, Ming Gao, Lilang Li, Yu Wang, Zhongsheng Lou, Juan Yang, Qiji Li, Xiaosheng Yang

**Affiliations:** 1School of Basic Medicine, Guizhou Medical University, Guiyang 550025, China; 2State Key Laboratory of Functions and Applications of Medicinal Plants, Guizhou Medical University, Guiyang 550014, China; 3Engineering Research Center of Natural Product Efficient Utilization in Guizhou, The Key Laboratory of Chemistry for Natural Products of Guizhou Province and Chinese Academy of Sciences, Guiyang 550014, China

**Keywords:** *Rosa roxburghii*, kaji-ichigoside F1, NMDA-induced neurotoxicity, neuroprotection, BDNF/AKT/mTOR

## Abstract

Kaji-ichigoside F1 (KF1), a natural oleanane-type triterpenoid saponin, is the main active constituent from *Rosa roxburghii*. In the southwest regions of China, particularly in Guizhou Province, this plant was used as a *Miao* ethnic medicine to prevent and treat dyspepsia, dysentery, hypoimmunity, and neurasthenia. In the present study, the neuroprotective effect of KF1 was evaluated against N-methyl-D-aspartate (NMDA)-induced neurotoxicity in vivo and in vitro. An NMDA-induced PC12 cell neurotoxicity assay showed that KF1 effectively improved cellular viability, inhibited the release of lactate dehydrogenase (LDH), and reduced cell apoptosis. Furthermore, KF1-treated NMDA-induced excitotoxicity mice displayed a remarkable capacity for improving spatial learning memory in the Y-maze and Morris water maze tests. In addition, KF1 increased the levels of the neurotransmitters 5-hydroxytryptamine, dopamine, and monoamine oxidase and reduced the calcium ion concentration in the hippocampus of mice. Hematoxylin and eosin and Nissl staining indicated that KF1 effectively reduced the impairment of neurons. Furthermore, Western blot assays showed that KF1 decreased NMDAR1 expression. In contrast, the NMDAR2B (NR2B), glutamate receptor (AMPA), TrkB, protein kinase B (AKT), mammalian target of rapamycin (mTOR), PSD95, and synapsin 1 were upregulated in NMDA-induced PC12 cells and an animal model. These results suggest that KF1 has a remarkable protective effect against NMDA-induced neurotoxicity, which is directly related to the regulation of the NMDA receptor and the activation of the α-amino-3-hydroxy-5-methylisoxazole-4-propionic acid receptor (AMPAR) and BDNF/AKT/mTOR signaling pathways.

## 1. Introduction

Neurotoxicity is an important contributor to central nervous system dysfunction, which is manifested as neuronal dysfunction, synaptic reduction, and cell death, and occurs in conditions such as depression and neurodegenerative diseases [1]. Many factors can induce neurotoxicity in serious neurodegenerative disorders, such as glutamate, methamphetamine, metallic ions, parasites, antipsychotic drugs, environmental pollutants, and food additives [2,3]. Glutamate is one of the most familiar neurotoxins that induces neurotoxic or excitotoxic cascades in different pathophysiological events and is associated with an influx of intracellular calcium ions (Ca^2+^) via N-methyl-D-aspartate receptor (NMDAR) activation [4]. Furthermore, activation of extrasynaptic NMDAR directly results in cell death [5]. It is noteworthy that diverse NMDAR GluN2 subunits have disparate physiological characteristics; GluN2A-containing NMDARs located at the brain synapses mediate in the cell survival pathway, but GluN2B-containing NMDARs located at extrasynaptic sites are involved in the cell death pathway [6,7]. Additionally, NMDA receptor subtype 2B (NR2B) overactivation led to an increased calcium concentration and excitotoxicity under pathological conditions [8]. These activation processes play a key role in synaptic transmission, learning and memory, and brain development [9]. Hence, NMDA-induced neurotoxicity evaluations can be used to discover neuroprotective drugs.

It is well known that a decrease in α-amino-3-hydroxy-5-methyl-4-isoxazolepropionic acid (AMPA) expression leads to reduced synaptic plasticity and learning and memory deficits and has been shown to be closely associated with depression and neuropsychiatric disorders [10]. Positive AMPA receptor modulators further increased brain-derived neurotrophic factor (BDNF) protein levels in vitro and in vivo, which influenced the function of neurons, synapses, and the central nervous system [11]. BDNF protein overexpression can activate tyrosine kinase receptor B (TrkB) receptors and the downstream PI3K/Akt/mTOR signaling pathway, which directly upregulates the expression of synaptic signaling proteins [12]. The mechanism of these signal proteins in neuroprotection has also been verified in previous reports and provides an effective method for discovering neuroprotective drugs.

*Rosa roxburghii* is a well-known medicinal and edible plant in China which is widely cultivated in Guizhou Province [13]. The earliest records of the plant come from the Book of Qian (1690 A.D.) and Compendium of Materia Medica (1765 A.D.). Today, this plant is used to prevent and treat dyspepsia, neurasthenia, sleep disorders, dysentery, and aging effects [14,15]. Our previous study confirmed that the total triterpenoids from *R. roxburghii* have remarkable antidepressant effects and that the concentration of KF1 surpasses 0.17 mg/g [16,17]. Therefore, this study further investigated the neuroprotective effects of KF1 on NMDA-induced excitotoxicity in vitro and in vivo. The results showed that KF1 has a remarkable protective effect against NMDA-induced neurotoxicity which is directly associated with the BDNF/AKT/mTOR signaling pathways.

## 2. Results

### 2.1. The Neuroprotective Mechanism of KF1 in PC12 Cells

#### 2.1.1. KF1 Alleviated NMDA-Induced Neurotoxicity in PC12 Cells

The neurotoxicity of KF1 was evaluated using different concentrations (from 5 μM to 80 μM); the results showed that KF1 does not significantly affect cell viability (Figure 1A). Furthermore, NMDA decreases cell viability in a concentration-dependent manner, as shown in Figure 1B. Compared with the vehicle group, the cell survival of the NMDA-treated group at 6 mM was about 55% after 6 h. Interestingly, when we added different doses of KF1 to NMDA-treated cells, the cellular viability significantly increased to 79%, 82%, and 89% (Figure 1C), and the lactate dehydrogenase (LDH) release decreased to 45%, 39%, and 28% (Figure 1D), respectively.

#### 2.1.2. Effects of KF1 on NMDA-Induced Ca^2+^ Overload

Next, we assessed the regulation of NMDA-induced Ca^2+^ in PC12 cells by KF1. As shown in Figure 2A,D, the NMDA-induced group had significantly increased intracellular Ca^2+^ levels in contrast to the vehicle group (*p* < 0.01), but KF1 clearly decreased the Ca^2+^ levels in comparison with the NMDA group (*p* < 0.01).

#### 2.1.3. Effects of KF1 on NMDA-Induced Apoptosis in PC12 Cells

Hoechst 33342 staining (Solar Bio, China) and annexin V/propidium iodide (PI) flow cytometry (Meilun Bio, China) were used to evaluate NMDA-induced apoptosis in PC12 cells. In contrast with the shrinkage of nuclei and the condensation of chromatin in the NMDA-induced group, KF1 treatment inhibited these apoptotic characteristics (Figure 2B). The NMDA-induced group experienced an increase in visibly necrotic cells in the Q2-2 quadrant (annexin V+/PI+) and apoptotic cells in the Q2-4 quadrant (annexin V+/PI−) compared with the vehicle group (*p* < 0.01), but the KF1-treated group had a visibly reduced percentage of apoptotic cells (*p* < 0.01, Figure 2C,E).

#### 2.1.4. KF1 Effects on the Expression of NR2B, AMPA, BDNF, AKT, mTOR, PSD95, and Synapsin 1 (Syn1) Proteins in NMDA-Induced PC12 Cells by Immunofluorescence

The expression of NR2B, AMPA, BDNF, AKT, mTOR, PSD95, and Syn1 proteins in NMDA-induced PC12 cells treated with KF1 was determined by immunofluorescence (Figure 3). The mean fluorescence intensities of NR2B (1:100 dilution, Abcam, Cambridge, UK), AMPA (1:100 dilution, CST, USA), BDNF (1:100 dilution, Abcam, USA), AKT (1:100 dilution, CST, USA), mTOR (1:100 dilution, CST, USA), PSD95 (1:100 dilution, Abcam, USA), and Syn1 (1:100 dilution, CST, USA) in the NMDA-induced group decreased significantly compared with in the vehicle group (*p* < 0.01). The fluorescence intensity of the KF1-treated groups and dizocilpine (MK-801)-treated group increased significantly (*p* < 0.01) compared with the NMDA-induced group.

### 2.2. In Vivo Analysis of the Mechanisms of Action of KF1

#### 2.2.1. Effect of KF1 on Spatial Memory in the Y-Maze Test

The Y-maze test was performed to evaluate the spatial memory of KF1-treated mice (Figure 4A,B). Compared with the vehicle group, the novel arm region exploration numbers, travel distance, and total duration were inferior in the NMDA-induced group (*p* < 0.05), which indicated poor spatial exploration abilities in NMDA-induced mice. After 14 days of treatment with MK-801 or KF1, the spatial memory of KF1-treated mice was improved. The KF1 treatment groups had increased novel arm region of exploration numbers, travel distance, and total duration (Figure 4C–E; *p* < 0.05) compared with the NMDA-induced group.

#### 2.2.2. Effect of KF1 on the Morris Water Maze (MWM) Test

The spatial learning and memory of KF1-treated mice were analyzed using the MWM test. In this study, the escape latency of the NMDA-induced group was significantly higher than that of the vehicle group (*p* < 0.05), which indicated that NMDA seriously damages the cognitive ability of the mice (Figure 5). Nevertheless, after KF1 treatment, the cognitive impairment was improved, especially in mice receiving KF1 treatment from Day 2 to Day 4 in different doses (Figure 5C; *p* < 0.05). On the fifth day, the NMDA-induced group spent less time than the vehicle mice in the target quadrant (*p* < 0.05). The KF1-treated group had increased retention time in the target quadrant compared to in the other quadrants (Figure 5D; *p* < 0.05). These data support that KF1-teated mice have improved spatial learning and memory.

#### 2.2.3. Effects of KF1 on Neurotransmitters and Monoamine Oxidase (MAO) Levels in the Hippocampus

The levels of the neurotransmitters 5-hydroxytryptamine (5-HT), dopamine (DA), and monoamine oxidase (MAO) in the hippocampus were examined by enzyme-linked immunosorbent assay (ELISA) (Figure 6). The levels of 5-HT and DA were decreased in the hippocampus of the NMDA-induced group compared with in the hippocampus of the vehicle group. However, the KF1-treated group had increased levels of the two neurotransmitters (Figure 6A,B; *p* < 0.05). In contrast, the level of MAO increased significantly in the hippocampus of the NMDA-induced group compared with in the hippocampus of vehicle mice (*p <* 0.05). After two weeks of KF1 treatment, the level of MAO was inhibited (*p <* 0.05).

#### 2.2.4. Effects of KF1 on Ca^2+^ Concentration

The hippocampus Ca^2+^ concentration was measured by commercial ELISA kit, and the results are shown in Figure 6D. Compared with the vehicle group, the hippocampus Ca^2+^ concentration increased in the NMDA-induced group (*p* < 0.05). In contrast, the hippocampus Ca^2+^ concentration of the KF1-treated group decreased significantly.

#### 2.2.5. Histopathological Changes in the Hippocampus

Hippocampus histopathological changes were observed by using hemotoxin and eosin (HE) staining. The hippocampal neurons were structurally intact, had good arrangement, and were round in the vehicle group (Figure 7A). In the NMDA-induced group, the neurons appeared with pyknotic nuclei, an irregular arrangement, deepened staining, and karyopyknosis. In contrast, the KF1-treated groups inhibited the NMDA-induced histopathological damage and exhibited intact neurons.

#### 2.2.6. Effects of KF1 on Hippocampal Neurons of NMDA-Induced Mice

Nissl staining was used to evaluate pyramidal cell neuronal injury in the hippocampus CA1 region. The pyramidal cells exhibited normal morphology, without nuclear pyknosis, and were arranged neatly in the vehicle group (Figure 7B,C). However, Nissl bodies were overtly reduced, and the neurons appeared abnormal with nuclear atrophy in the hippocampus CA1 region of the NMDA-induced group. The KF1-treated groups had an increased number of Nissl bodies compared with the NMDA-induced group (*p* < 0.01).

#### 2.2.7. Immunohistochemical Expression of NMDAR1 in the Hippocampus and Cortex

The immunocytochemical data demonstrated that NMDAR1 was expressed in the CA1, CA3, and cortex regions (Figure 7D–I), and the mean optical density value of NMDAR1 was increased markedly in the hippocampus and cortex of the NDMA-induced group compared with in those of the vehicle group (*p* < 0.01). In the KF1-treated groups, the mean optical density of NMDAR1 decreased noticeably in the CA1, CA3, and cortex regions (*p <* 0.01).

#### 2.2.8. Western Blot Assay to Verify the Expression of NMDAR1, AMPA, BDNF, TrkB, AKT, mTOR, PSD95, and Syn1 In Vivo and In Vitro

The neuroprotective mechanism of KF1 was also confirmed by using Western blot assays to verify the expression of interacting proteins and signaling pathways in vivo and in vitro. Compared with the vehicle group, the expression of AMPA, BDNF, AKT, mTOR, PSD95, and Syn1 (*p* < 0.05) was inhibited in NMDA-induced PC12 cells (Figure 8). Interestingly, the expression of NR2B, AMPA, BDNF, TrkB, AKT, mTOR, PSD95, and Syn1 (*p* < 0.05) was also inhibited in the hippocampus of NMDA-induced mice (Figure 9). Furthermore, NMDA caused overexpression of NMDAR1 in PC12 cells and the hippocampus (*p* < 0.05). However, the increased protein expression in the NMDA group was reversed in the KF1- and MK-801-treated groups (*p* < 0.05). Hence, these interacting proteins and signaling pathways play a key role in the KF1-mediated neuroprotective mechanism.

## 3. Discussion

It is well known that the hippocampus is the most important tissue organ regulating cognition, emotion, and learning memory. The regulation of proteins, receptors, and neurotransmitters associated with the hippocampus is a key mechanism for neuroprotective effects in vitro and in vivo. This study demonstrated the neuroprotective effect of KF1 against NMDA-induced neurotoxicity in vivo and in vitro. First, KF1 attenuated NMDA-induced dose-dependent neurotoxicity by increasing cell viability and reducing intracellular Ca^2+^ concentrations, apoptosis, and LDH release in PC12 cells. Second, the MWM navigation task, which has been widely used to study spatial learning and memory in behavioral neuroscience for animals, and the Y-maze behavioral tools were used to conduct cognitive assessments on rodents. Both tests are specialized for cognitive evaluation in behavioral neuroscience in rodents and are extensively applied [18,19]. The MWM results of the present study showed that learning and memory were impaired in the NMDA-induced group, while KF1 treatment improved the learning and memory abilities of mice. In the Y-maze test, the NMDA-induced group had a decreased ability to explore the novel environment; in contrast, KF1 treatment reversed these results, showing an increase in the number of novel arm entries, the duration in the novel arm, and the distance traveled in the novel components. Previous studies have shown that NMDA-induced neurotoxicity can be assessed by the MWM and Y-maze behavioral testing. The MWM and Y-maze assays showed that KF1 improved spatial learning and memory in mice, and these behavioral results are consistent with the literature [20,21]. MAO is one of the main metabolic enzymes of DA metabolism and catalyzes 5-HT and 5-HT receptors to regulate glutamate-induced neurotoxicity through multiple synaptic circuits related to the glutamate system [22]. In this study, the neuroprotective effect of KF1 against NMDA-induced neurotoxicity was further supported by an increase in 5-HT and DA and a decrease in MAO, whereas KF1 treatment reversed this trend. Previous studies were consistent with our experimental results [23,24]. Third, the hippocampus is the central target of the limbic system in the brain, which is associated with various forms of cognition, emotion, and learning memory [25]. HE and Nissl staining are crucial methods used to observe hippocampus morphological changes, and impaired neuronal cores in the CA1 region exhibited irregular deformation, atrophy, and uneven or deepened cytoplasmic staining [25,26]. Fortunately, KF1 significantly reduced the percentage of impaired neuronal cores and cell loss in the CA1 region. In excitatory synapses in the CNS, the increase of Ca^2+^ by NMDAR activation is a crucial factor resulting in NMDA-induced neurotoxicity, but an AMPA-induced increase of Ca^2+^ did not show neurotoxicity [27,28]. Interestingly, in the present study, our data showed that Ca^2+^ was significantly increased in the NMDA group compared with the vehicle group, and the KF1-treated groups repressed the NMDA-induced Ca^2+^ increase in vivo and in vitro; these results were confirmed in previous studies [29]. As a result, KF1 was found to inhibit NMDA-induced Ca^2+^ in vivo and in vitro.

NMDAR and AMPAR are considered as important types of ionic glutamate receptors, and their expression is activated by glutamate from the prefrontal cortex, hippocampus, and other brain regions [30]. In immunohistochemistry and Western blot assays, the NMDAR1 expression of the NMDA-induced group was increased in vivo and in vitro, but the expression of AMPA and NR2B was reduced. Conversely, the KF1-treated groups displayed an increase in NMDAR1 expression and a decrease in AMPA and NR2B expression. These results are similar to previous findings and further confirm that NMDAR and AMPAR are relevant to NMDA-induced neurotoxicity [28].

BDNF is widely present in different circuits in the brain and plays a critical role in neuronal survival, synaptic plasticity, and memory [31]. TrkB is an endogenous receptor with a high affinity for BDNF and can activate the downstream signaling of the BDNF/TrkB axis through NMDAR, such as Akt/mTOR. BDNF binding to TrkB reduces symptoms in many pathological diseases, and overexpressed TrkB can improve spatial learning and memory in mice [32]. In the present study, we showed that expressions of BDNF and TrkB were decreased after NMDA-induced neurotoxicity in vivo and in vitro, whereas treatment with KF1 upregulated BDNF and TrkB expression compared with the NMDA group. Previous literature was consistent with our data [33]. Moreover, BDNF/TrkB-mediated signaling pathways activated mTORC1, which regulates synaptic plasticity to establish the basis of hippocampal-dependent learning and memory, and synthesis of synaptic proteins, such as PSD95 and synapsin 1, which are the foundation of modulating cell proliferation and survival [34,35]. In this study, the overexpression of BDNF and TrkB was observed in vivo and in vitro, and KF1 upregulated mTOR, PSD95, and synapsin 1 protein expression, which further supports that the BDNF/TrkB signaling pathways play a critical role in NMDA-induced excitotoxicity.

In addition, the AKT/mTOR signaling pathway is considered as the target of inhibitory neurotoxicity, and decreased Akt and mTOR protein expression causes neuronal apoptosis. Rui Li et al. revealed the neuroprotective effect of the Akt/mTOR signaling pathway activated after NMDA-induced injury [36,37]. In the synaptic region of neurons, mTOR is closely associated with neuronal survival and plasticity and controlled protein translation via the Akt pathway. This ultimately influences cell growth, proliferation, survival, and protein synthesis, which is further maintained by intracellular homeostasis [38]. In this experiment, KF1 inhibited NMDA-induced neurotoxicity by decreasing the protein levels of AKT and mTOR in PC12 cells and the hippocampus. The results showed that the AKT/mTOR pathway may be used to study neuroprotective effects against NMDA-induced neurotoxicity.

Natural products have always been an important source of drug discovery. As with a high number of natural products derived from medicinal and edible plants, KF1 is non-toxic, safe, and has other advantages for the development of neuroprotective drugs. Of course, a highly effective and safe drug still needs to be verified by several cell or animal models. In this research, we revealed the neuroprotection of KF1 against NMDA-induced neurotoxicity in vivo and in vitro. KF1 not only plays a neuroprotective role through its effects on the AKT/mTOR signaling pathway, but also targets NMDAR and AMPAR receptors. This new information provides a basis for further research on the application of KF1.

## 4. Materials and Methods

### 4.1. In Vitro Analysis of the Mechanisms of Action of KF1

#### 4.1.1. Cell Management and Viability

PC12 cells were obtained from the Institute of Biochemistry Cell Biology (Shanghai, China). Cells were cultured in a Dulbecco’s modified Eagle medium (DMEM) containing 10% fatal bovine serum (FBS), 100 U/mL streptomycin, and 100 U/mL of penicillin and placed in a 5% CO2 incubator at 37 °C. To study the neuroprotective effect of KF1, the experiment was divided into deferent groups: vehicle, in which cells were incubated in DMEM for 24 h; NMDA (Med Chem Express, USA), in which cells were cultured for 24 h in DMEM before adding 10 µM NMDA; and the NMDA plus MK-801 or KF1 treatment groups, in which PC12 cells were incubated in DMEM for 24 h and then treated with different concentrations of KF1 (5 μM, 10 μM, or 20 μM) or MK-801 (Med Chem Express, USA; 10 μM) for 24 h. NMDA (6 mM) was added 6 h before treatment with KF1 or MK-801 to assess the protective effect against NMDA-induced neurotoxicity [39]. Cell viability was evaluated using the MTT assay as previously described [40]. Cell viability was expressed by the percentage of absorbance of each group to the control group.

#### 4.1.2. LDH Release Assay

The LDH assay was performed as previously described [41]. The LDH (Beyotime, China) assay was used to determine the effects of KF1 on NMDA-induced cytotoxicity. Following treatment, supernatant from PC12 cells was analyzed by detecting LDH concentrations according to the kit instructions. LDH activity was measured at 450 nm by a microplate reader (Victor Nivo, PerkinElmer, USA).

#### 4.1.3. Measurement of Intracellular Ca^2+^ Concentration

The intracellular Ca^2+^ concentration was measured as previously described [42]. Briefly, PC12 cells were seeded on 6-well plates at a density of 1 × 10^5^ cells/well. At the end of the treatment, cells were incubated with 10 μM fluo-3/AM (Solarbio, China) for 40 min at 37 °C in the dark. Subsequently, the cells were resuspended with HEPES-buffered saline, and intracellular Ca^2+^ concentrations were determined by excitation and emission wavelengths at 506 nm and 526 nm, respectively. Furthermore, the fluorescence intensity of Ca^2+^ in PC12 cells was measured by laser scanning fluorescence microscopy.

#### 4.1.4. Hoechst 33342 Staining

To distinguish apoptotic from normal cells, Hoechst 33342 staining (Solarbio, China) was used for the qualitative analyses of PC12 cells [43]. Briefly, after treatment, cells are incubated with Hoechst 33342 for 15 min at room temperature, rinsed three times with phosphate-buffered saline (PBS), and then captured under an inverted fluorescence microscopy (Leica Microsystems, Wetzlar, Germany).

#### 4.1.5. Annexin V-FITC/PI Flow Cytometry Assay

This assay was performed as previously described [44]. Briefly, PC12 cells were cultured in 6-well plates, and, after drug treatment, the cells were resuspended and incubated for 15 min in the dark at room temperature. Apoptosis rates were analyzed by a Novocyte Flow Cytometer (NovoCyte D2040R, ACEA Biosciences, San Diego, CA, USA).

#### 4.1.6. Fluorescence Staining

To evaluate the protective effects of KF1 in NMDA-induced PC12 cells, cells were fixed with 4% paraformaldehyde for 20 min at room temperature and then rinsed with PBS three times (5 min each time), permeabilized with 0.3% Triton X-100 for 10 min, and blocked with normal goat serum for 1 h. Cell were incubated at 4 ℃ overnight with primary antibodies. Cells were then stained with secondary antibodies for 2 h at 37 °C, washed with PBS, and then dyed with 4′,6-diamidino2-phenylindole (DAPI). Photos were taken using a fluorescence microscope (Leica DMi8, Germany), and the mean intensity was determined by using ImageJ software [45].

### 4.2. In Vivo Analysis of the Mechanisms of Action of KF1

#### 4.2.1. Animals

Male Kunming mice (20 ± 22 g) were used for these studies and were purchased from Chongqing Tengxin Biotechnology Co., Ltd. (Chongqing, China, certificate no. SCXK (Liao) 2015-0001). All experiments were conducted according to the requirements of the Animal Care and Use Committee of Guizhou Medical University. The mice were housed under environmental conditions (12 h light–dark cycle, 55% humidity, 25 ± 2 °C) with ad libitum access to water and food.

#### 4.2.2. Experimental Designs

All mice were adaptively fed for 1 week. Seventy-two mice were randomly divided into six parallel groups (*n* = 12), including the vehicle group, NMDA group, NMDA + MK-801 (2.0 mg/kg) group, and NMDA + KF1 (1.0, 2.0, or 4.0 mg/kg) groups. NMDA-induced neurotoxicity methods were carried out as Boeck described with minor modifications [46]. Briefly, neurotoxicity was induced by NMDA (75 mg/kg; 10 mL/kg; i.p) in all mice except those in the vehicle (saline 0.9%; 10 mL/kg) group for 7 days. After 7 days, the vehicle group was not subjected to treatment and received the same volume of 0.5% (*w*/*v*) carboxymethyl cellulose sodium (CMC-Na) as the KF1 and MK-801 group, which was administered with treatment drugs once daily for two consecutive weeks. After 2 weeks, all mice underwent Y-maze and MWM analyses to evaluate the impact of KF1 on spatial learning and memory abilities. All mice were anesthetized by inhalation of diethyl ether. Three mouse brains were fixed for histology, and the remaining brains were flash frozen for Western blot and biological analyses.

#### 4.2.3. Y-Maze Test

The Y-maze test was performed as previously described [47]. Working memory and spontaneous spatial recognition were measured using a Y-maze apparatus which consisted of three arms (start, novel, and another arm) at 120-degree angles to each other. Mice were subjected to two-trial Y-maze tests. In the first trial of the test, the novel arm was blocked with a door, and mice were allowed to move freely between the other two compartments for 15 min. The second stage test was conducted after a 6 h intertrial interval, and animals were allowed free access to all three compartments for 5 min. Data were collected by an EthoVision automated tracking system to analyze the time spent exploring the familiar and novel arms and the total distance traveled during the 5 min period of the second trial.

#### 4.2.4. Morris Water Maze Test

The MWM was performed as previously described [48]. Briefly, this experiment aimed to assess the role of KF1 in the spatial learning and memory ability of mice. The MWM consisted of a circular (1.4 m diameter, 0.5 m high) pool filled with white water (22 °C). The circular surface was divided into four quadrants, with a hidden 12 cm diameter escape platform located 1 cm below the surface of the water in the center of the target quadrant. The day before training commenced, mice were allowed to explore freely for 120 s to acclimate to the MWM environment. In this task, animals received eight training sessions (two sessions/day, 90 s cutoff at every turn) for four consecutive days, and the probe trial was performed on the fifth day. The evaluation began when the mice were placed into the water facing the wall at one of the three starting positions. The animals were then permitted to swim from the start location to reach the escape platform (escape latency), and the information was recorded. On the fifth day, the escape platform was removed, and a spatial probe test was carried out. The data of each trial were recorded by a computer-based video tracking system (Taming Co., Chengdu, China).

#### 4.2.5. Quantification of 5-HT, DA, MAO, and Ca^2+^ Concentration in the Hippocampus by ELISA

Blood samples were collected by aorta ventralis puncture after the behavioral tests, the brains were rapidly removed, and the cortex and hippocampus were isolated. The samples were stored at −80 °C before assaying the expression levels of 5-HT, DA, MAO, and Ca^2+^ in the hippocampus and were measured by ELISA kits (Cusabio Technology Co., Ltd., Wuhan, China) according to the manufacturer’s instructions [49].

#### 4.2.6. HE Staining of the Hippocampus

Animals were sacrificed after the behavior tests, and brains were removed from three mice from each group and fixed in 4% paraformaldehyde. Next, the tissue was embedded in paraffin, and hippocampal tissue was stained with HE for assessing the damage to hippocampal neurons [49].

#### 4.2.7. Nissl Staining of the Hippocampus

For the experiment with Nissl staining, animals randomly selected from each group were sacrificed, and the left hippocampal tissues was fixed in 4% paraformaldehyde, embedded in paraffin, and sliced into 5 μm thick coronal sections according to a previously described method [48]. Xylene was used to deparaffinize the tissue slices, and an alcohol gradient was used for dehydrating the slices. Then, the tissue slices were incubated with Nissl staining solution (Solarbio, Beijing, China) for 5 min and finally sealed with a neutral fixative.

#### 4.2.8. Immunohistochemistry Assay for NMDAR1 in the Hippocampus and Cortex

Immunohistochemical staining was conducted as previously reported [50]. In brief, the whole mouse brains were fixed in 4% paraformaldehyde and embedded in paraffin, sliced into 5 µm thick slices and deparaffinized in xylene, and dehydrated in an ethanol gradient. Tissue was microwave treated for 3 min, blocked with 3% H_2_O_2_, and incubated in 5% normal goat serum. The tissue slices were incubated overnight with a primary antibody against NMDAR1 at 4 °C, then incubated with a biotinylated secondary antibody and avidin–biotin horseradish peroxidase complex for 30 min at room temperature. Immediately, serial tissue sections of each sample were stained with a DAB solution and sealed with a neutral gum. The immunostained tissues were evaluated under a microscope (KS300, Zeiss-Kontron, Munich, Germany).

#### 4.2.9. Western Blot

Western blot assays were conducted as previously reported [51]. Briefly, the hippocampus of mice was sheared into small pieces, and the total protein of the hippocampus and PC12 cells was incubated with RIPA lysis buffer on ice. The supernatant was collected, and the protein concentration was measured by a BCA protein assay kit (Solarbio, China). The protein was separated by SDS-PAGE gel electrophoresis according to relative molecular weight and then proteins were transferred onto polyvinylidene fluoride (PVDF) membranes (Millipore Trading Co., Ltd., Shanghai, China). The membranes were blocked with 5% nonfat milk for 2 h at room temperature and then incubated in the corresponding primary antibodies overnight at 4 °C. After 16–18 h, the PVDF membranes were incubated with horseradish peroxidase conjugated secondary antibodies at 37 °C for 2 h. After being washed, the protein bands were detected using an enhanced chemiluminescence (ECL) select kit (CodeNo. 16929851), visualized using gel imaging (Bio-Rad, USA), and quantified using ImageJ software.

### 4.3. Statistical Analysis

All experiments were repeated at least three times, and values were expressed as mean ± SEM. All results were analyzed by GraphPad Prism 5.0 (GraphPad Software, La Jolla, CA, USA). Statistical comparisons between experimental groups and the vehicle group were performed by using one-way analysis of variance (ANOVA) followed by Tukey’s post hoc test for comparisons between multiple groups. *p* < 0.05 was considered statistically significant.

## 5. Conclusions

In conclusion, the neuroprotective effect of KF1 against NMDA-induced neurotoxicity was verified in vivo and in vitro. On the basis of dose-dependent NMDA-induced intracellular Ca^2+^ overload in vivo and in vitro, we found that KF1 increased cell viability and decreased apoptosis, improved learning and memory functions, and increased neuronal survival. In terms of the mechanisms, KF1 significantly improved NMDA-induced neurotoxicity by regulation of NMDAR and activation of AMPAR and the BDNF/AKT/mTOR signaling pathways. In brief, KF1 is an excellent natural inhibitor against NMDA-induced neurotoxicity, with potential as a neuroprotective drug.

## Figures and Tables

**Figure 1 ijms-23-16150-f001:**
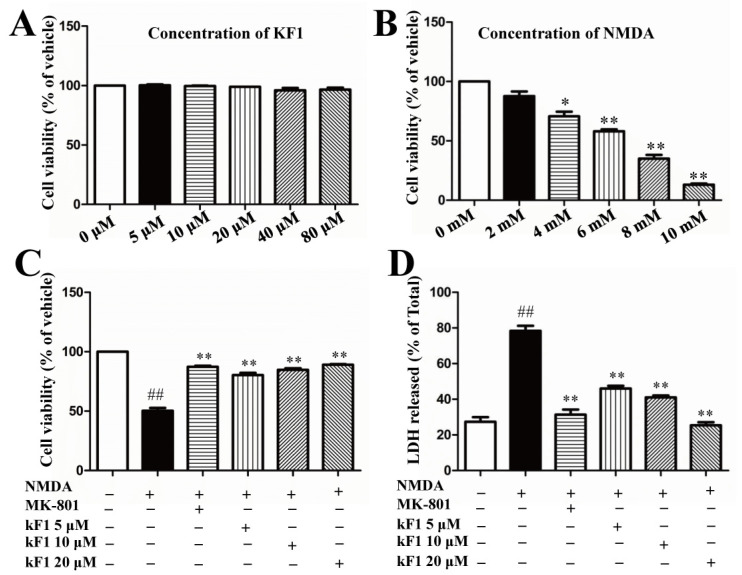
KF1 inhibited NMDA-induced neurotoxicity. (**A**) The cytotoxicity of KF1 in 24 h. The data represent mean ± SEM (*n* = 10). (**B**) The cell viability of NMDA-treated cells after 6 h; * *p* < 0.05; ** *p* < 0.01 versus 0 mM group. (**C**) The protection of KF1 against NMDA-induced neurotoxicity. ^##^
*p* < 0.01 versus vehicle group; ** *p* < 0.01 versus NMDA group. (**D**) The release of lactate dehydrogenase (LDH). The data represent mean ± SEM (*n* = 8). ^##^
*p* < 0.01 versus vehicle group; ** *p* < 0.01 versus NMDA group.

**Figure 2 ijms-23-16150-f002:**
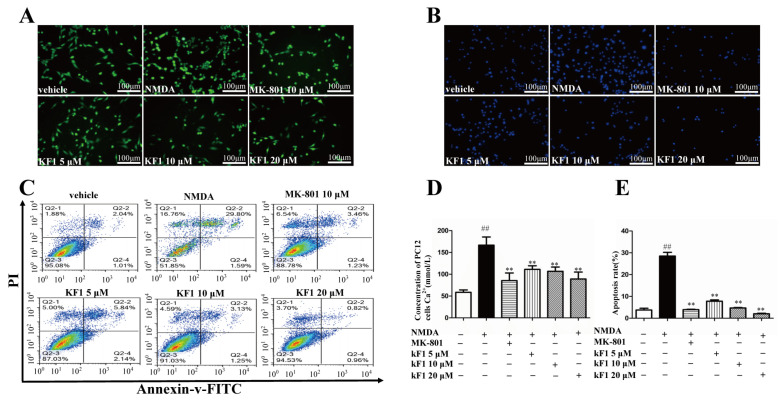
Effects of KF1 on NMDA-induced neurotoxicity in PC12 cells. (**A**) Fluorescence images of intracellular calcium in different groups. (**B**) Representative fluorescence images of Hoechst 33342 staining. (**C**) Effects of KF1 on NMDA-induced early and late apoptosis after AV/PI double staining. (**D**) Effect of KF1 on NMDA-induced Ca^2+^ increase in PC12 cells. (**E**) The quantitative analysis of early and late apoptotic cells. Bars: 100 μm, the data represent mean ± SEM (*n* = 5). ^##^ *p* < 0.01 versus vehicle group; ** *p* < 0.01 versus NMDA group.

**Figure 3 ijms-23-16150-f003:**
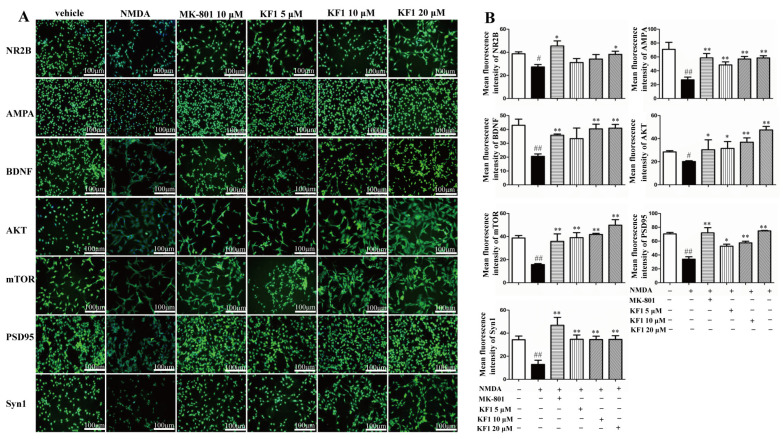
The immunofluorescence of NR2B, AMPA, BDNF, AKT, mTOR, PSD95, and Syn1 in PC12 cells. (**A**) Fluorescence images. (**B**) The mean fluorescence intensity of NR2B, ^#^ *p* < 0.01 versus vehicle group, * *p* < 0.01 versus NMDA group; AMPA, ^##^ *p* < 0.01 versus vehicle group, * *p* < 0.01 versus NMDA group; BDNF, ^##^ *p* < 0.01 versus vehicle group, * *p* < 0.01 versus NMDA group; AKT, ^#^ *p* < 0.01 versus vehicle group, * *p* < 0.05, ** *p* < 0.01 versus NMDA group; mTOR, ^##^ *p* < 0.01 versus vehicle group, ** *p* < 0.01 versus NMDA group; PSD95, ^##^ *p* < 0.01 versus vehicle group, * *p* < 0.05, ** *p* < 0.01 versus NMDA group; and Syn1 ^##^ *p* < 0.01 versus vehicle group, ** *p* < 0.01 versus NMDA group;. Green fluorescence represents fluorescent staining, and blue fluorescence represents DAPI-stained nuclei. Bars: 100 μm, the data represent mean ± SEM (*n* = 5).

**Figure 4 ijms-23-16150-f004:**
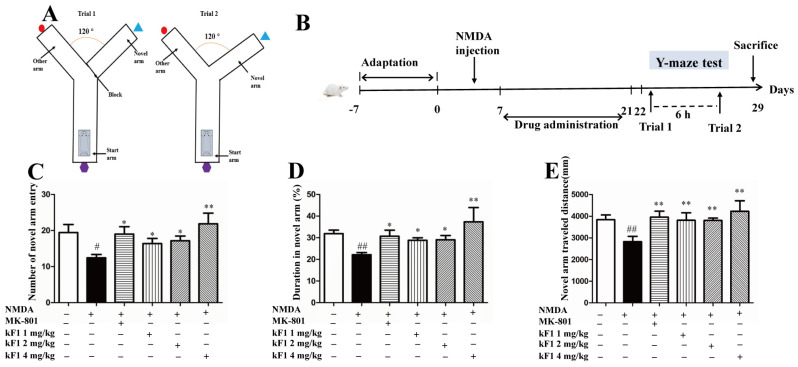
Effect of KF1 on spatial memory in the Y-maze test. (**A**) The two-trial Y-maze apparatus. (**B**) Experimental procedure. (**C**) The novel arm region of exploration numbers. (**D**) The novel arm region of total duration rate. (**E**) The novel arm region of travel distance. All data represent the mean ± SEM. ^#^ *p* < 0.05, ^##^ *p* < 0.01 versus vehicle group; * *p* < 0.05, ** *p* < 0.01 versus NMDA group (each group, *n* =10-12).

**Figure 5 ijms-23-16150-f005:**
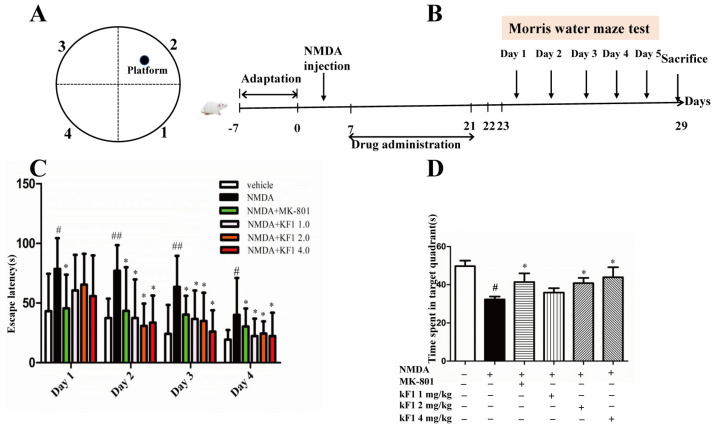
The spatial learning and memory of KF1 -treated mice in the MWM test. (**A**) The MWM schematic. (**B**) The MWM test design. (**C**) The latencies after 4 days of training trials. The data are expressed as means ± SEM (*n* = 10–12); ^#^
*p* < 0.05, ^##^
*p* < 0.01 versus vehicle group; * *p* < 0.05 versus NMDA group. (**D**) The retention time in the target quadrant. The data are expressed as means ± SEM (*n* = 10–12); ^#^ *p* < 0.05 versus vehicle group; * *p* < 0.05 versus NMDA group.

**Figure 6 ijms-23-16150-f006:**
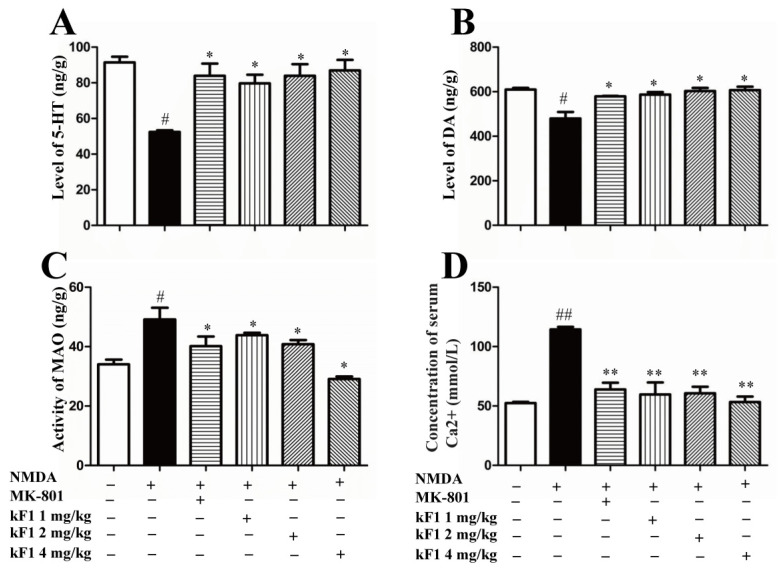
Effects of KF1 on neurotransmitters, MAO levels, and Ca^2+^ concentration in the hippocampus. The levels of 5-HT (**A**) and DA (**B**); the activities of MAO (**C**) and Ca^2+^ (**D**) concentration in the hippocampus of NMDA-induced mice. The data are expressed as means ± SEM (*n* = 8); ^#^ *p* < 0.05, ^##^ *p* < 0.01 versus vehicle group; * *p* < 0.05, ** *p* < 0.01 versus NMDA group.

**Figure 7 ijms-23-16150-f007:**
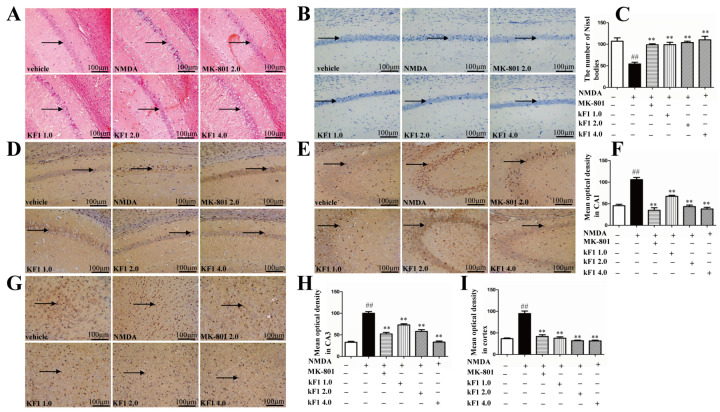
Histopathological changes in the mouse brain. (**A**) Hippocampus coronal paraffin sections; (**B**) representative Nissl bodies in the hippocampal; (**C**) quantitation of pyramidal cells in the hippocampal CA1; (**D**) NMDAR1 expression level in the hippocampus CA1; (**E**) NMDAR1 expression level in the hippocampus CA3; (**G**) NMDAR1 expression level in the cortex. (**F**,**H**,**I**) Immunohistochemistry mean optical density (MOD) was used to calculate NMDAR1 expression levels in each group. Bars: 100 μm; the data represent mean ± SEM (*n* = 3). ^##^ *p* < 0.01 versus vehicle group; ** *p* < 0.01 versus NMDA group.

**Figure 8 ijms-23-16150-f008:**
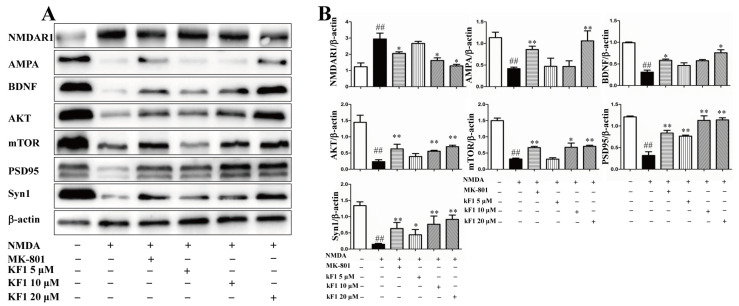
Effect of KF1 on the expression of related proteins in the PC12 cells. (**A**) Related protein expression; (**B**) quantitation of interacting protein levels in the PC12. The relative optical density was normalized to β-actin. The data are expressed as means ± SEM (*n* = 3). ^##^ *p* < 0.01 versus vehicle group; * *p* < 0.05, ** *p* < 0.01 versus NMDA group.

**Figure 9 ijms-23-16150-f009:**
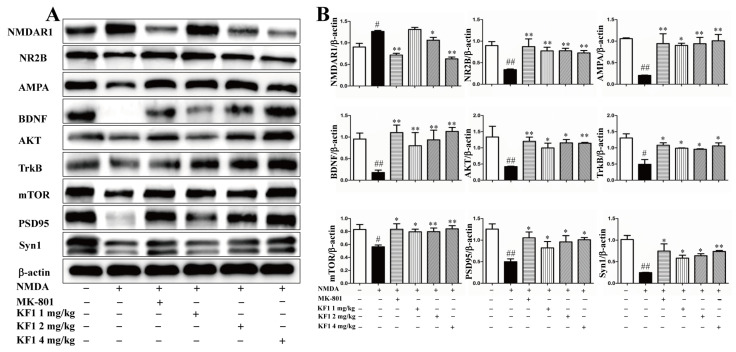
Effect of KF1 on the expression of related proteins in hippocampus. (**A**) Interacting protein expression in hippocampus; (**B**) quantitation of interacting protein levels in hippocampus. The relative optical density was normalized to *β*-actin. The data are expressed as means ± SEM (*n* = 3). ^#^ *p* < 0.05, ^##^ *p* < 0.01 versus vehicle group; * *p* < 0.05, ** *p* < 0.01 versus NMDA group.

## Data Availability

The data that support the findings of this study are available from the corresponding author upon reasonable request.

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
