# Peer review of "Neuroprotection of Kaji-Ichigoside F1 via the BDNF/Akt/mTOR Signaling Pathways against NMDA-Induced Neurotoxicity"

_ijms, 2022, doi:10.3390/ijms232416150_

Round 1

Reviewer 1 Report

The manuscript by Chen et al focuses on the effect of KF-1 derived from Rosa roxburghii on neuroprotection following NMDA mediated neurotoxicity. The current study demonstrates that application of Kf-1 ameliorates damages caused by NMDA by decreasing cell apoptosis, decreasing Nissl bodies, decreasing Ca+2 levels while at the same time increasing levels of neurotransmittters. Additionally, application of KF-1 ameliorates cognition in mice following NMDA treatment. Although the mechanism by which KF-1 protects from NMDA-mediated neurotoxicity is not clear from this study, the observations are interesting for general and a more specialized audience. However, there are several issues with this manuscript, mostly with how it is written:

1. The manuscript presents the observations as outcomes of seemingly unconnected experiments. However, the norm is to combine all the observations to form a coherent story where experiments follow a logical pattern. The authors need to extensively rewrite the manuscript keeping this in mind. Additionally, the current manuscript is replete with grammatical errors which often makes the sentences very difficult to understand. The authors need to fix this in the revised manuscript.

2. In figure 3, a proper quantitative method has not been used to compare levels of different proteins. The authors need to use quantitative tests to support their claim. At least the authors can use semi-quantitative tests like western blot to compare levels of different proteins between different conditions.

3. Lastly, the authors need to provide full forms for all abbreviations when they are first mentioned.

Reviewer 2 Report

The manuscript provides interesting results; however, some concerns regarding the methodologies should be clarified before further consideration

1. Regarding statistical analysis, the authors did not perform a post-hoc comparison tests to assess the differences. These test are important to be performed, otherwise the results will not be reliable. Please, clarification.  Subsequently, the obtained results should be clarified and revised.

2. The results should be more discussed and rationalized with more adequate previous studies.

3. Materials and Methods: Please, provide proper references for all performed methods. This point is very important to validate that the authors have conducted the research properly according to certified methods.

4. I recommend the authors double-check the full text for grammatical and typing errors.

Round 2

Reviewer 1 Report

The authors have addressed all my concerns satisfactorily.  I recommend the current version for publication.

Reviewer 2 Report

The manuscript has been sufficiently improved.